# Sugary Liquids in the Baby Bottle: Risk for Child Undernutrition and Severe Tooth Decay in Rural El Salvador

**DOI:** 10.3390/ijerph18010260

**Published:** 2020-12-31

**Authors:** Priyanka Achalu, Abhishek Bhatia, Bathsheba Turton, Lucy Luna, Karen Sokal-Gutierrez

**Affiliations:** 1School of Public Health, University of California, Berkeley, Berkeley, CA 94720, USA; ksokalg@berkeley.edu; 2FXB Center for Health and Human Rights, The Lakshmi Mittal and Family South Asia Institute, Harvard University, Cambridge, MA 02138, USA; abhi@mail.harvard.edu; 3Department of Dentistry, University of Puthisastra, Phnom Penh 12211, Cambodia; bethy.turton@gmail.com; 4Asociación Salvadoreña Pro Salud Rural (ASAPROSAR), Santa Ana 02201, El Salvador; direccionasaprosar2016@gmail.com

**Keywords:** child undernutrition, nutrition transition, artificial feeding, dental caries, sugar-sweetened beverages, risk factors, caregivers, El Salvador

## Abstract

As communities worldwide shift from consuming traditional diets to more processed snacks and sugar-sweetened beverages (SSBs), increases in child obesity and tooth decay and persistence of undernutrition are particularly apparent in Latin American countries. Further evidence of shared risk factors between child undernutrition and poor oral health outcomes is needed to structure more effective health interventions for children’s nutrition. This study aims to identify dietary, oral health, and sociodemographic risk factors for child undernutrition and severe early childhood caries (sECC) among a convenience sample of 797 caregiver–child pairs from rural Salvadoran communities. Caregiver interviews on child dietary and oral health practices were conducted, and their children’s height, weight, and dental exam data were collected. Multivariable regression analyses were performed using RStudio (version 1.0.143). Caregiver use of SSBs in the baby bottle was identified as a common significant risk factor for child undernutrition (*p* = 0.011) and sECC (*p* = 0.047). Early childhood caries (*p* = 0.023) was also a risk factor for developing undernutrition. Future maternal–child health and nutrition programs should coordinate with oral health interventions to discourage feeding children SSBs in the baby bottle and to advocate for policies limiting SSB marketing to young children and their families.

## 1. Introduction

The “nutrition transition”, characterized by a global shift from traditional, whole-food diets to foods and beverages that have high sugar, fat, and salt content, has been driven by global increases in urbanization, trade liberalization, and food marketing [1]. The consequences of this transition are felt mostly in low-to-middle-income countries, and over recent decades, Latin America specifically has seen an increased consumption of fat and sugar and decreased intake of fresh fruit and vegetables [1,2]. Prior research has highlighted the link between this shift in dietary patterns and malnutrition: the deficiency, excess, or imbalance in an individual’s consumption of energy and nutrients [3]. As the global nutrition landscape changes, the literature has especially noted the double burden of malnutrition, where the prevalence of obesity increases while undernutrition persists [4]. Young children who are still growing and developing are especially vulnerable to these nutrition-related diseases [4]. Children with malnutrition, which includes both overweight/obesity and undernutrition, are at higher risk for overall poorer health, development, and quality of life and for tooth decay [4,5].

Tooth decay (dental caries) is the most common chronic childhood disease worldwide, affecting 60–90% of children worldwide, and studies have also demonstrated that children with severe dental caries are at higher risk for poor nutrition outcomes [6,7,8]. While many factors contribute to the progression of dental caries such as diet, oral bacteria biofilm, poor oral hygiene, and socioeconomic conditions, research has recognized dietary sugars as the main driver of this disease [9,10]. Research has attributed the coexistence of malnutrition and early childhood caries (ECC)—the presence of one or more decayed teeth in a child under age 6—to a shared set of risk factors including frequent consumption of processed foods and sugar-sweetened beverages, and severe caries leading to chronic dental infection, inflammation and mouth pain, and subsequent malnutrition [7,11]. However, the mechanisms by which ECC negatively impacts a child’s nutrition are not fully understood [12,13].

El Salvador, like other low-to-middle-income countries, experiences the double burden of malnutrition [14,15]. While approximately 1 in 4 children living in rural El Salvador experience chronic undernutrition, obesity has become an emerging public health problem; 6% of all Salvadoran children under the age of 5 are overweight or obese, increasing to 49% by adulthood [4,16]. In El Salvador, over 50% of young children also experience ECC [17]. Because access to dental services is limited for much of the Salvadoran population, most cavities resulting from tooth decay remain untreated, which can cause mouth pain, difficulty eating and sleeping, poor school performance, and an overall worsened quality of life [8,18]. While existing studies have documented the effects of the nutrition transition on children’s nutrition status in broader Latin American contexts, its effects in rural El Salvador specifically remain unclear [19].

This study is an exploratory analysis aiming to identify dietary, oral health, and sociodemographic risk factors for child undernutrition and severe early childhood caries (sECC) for rural communities in El Salvador. 

## 2. Materials and Methods 

### 2.1. Study Population

The study population included a cross-sectional convenience sample of mothers and children in El Salvador who were beneficiaries of the local Salvadoran partner Asociación Salvadoreña Pro Salud Rural (Salvadoran Association for Rural Health, ASAPROSAR). ASAPROSAR is a not-for-profit, nongovernmental community health and development program that works with community health workers (*promotoras de salud*) to promote maternal and child health practices for low-income, low-literacy, and resource-poor communities in the rural Santa Ana region. ASAPROSAR *promotoras*, working in 15 rural communities, hosted oral health and nutrition camps with Ministry of Health dentists 1–2 times per year from July 2006 through July 2010, where they provided nutrition and oral health education, free-of-cost dental examinations, referral to further treatment, fluoride varnish applications, toothbrushes, and fluoride toothpaste. At these camps, *promotoras* verbally explained the study details and invited all families with children 6 months through 6 years of age to participate in the study. Children below the age of 6 months, and at or above the age of 6 years were excluded from the study population but were given nutrition and oral health education. Mothers provided their signature or thumbprint as written informed consent for their own and their child’s participation. Children provided verbal assent to participate according to their developmental ability. The study was approved by the University of California, Berkeley’s Institutional Review Board (#2010-06-1655) and ASAPROSAR directorship.

### 2.2. Data Collection

Data collected consisted of (1) the mother’s interview, (2) child anthropometric measures (height and weight), and (3) child dental examination. The survey tool (Appendix A) was adapted from the World Health Organization (WHO) Oral Health Surveys and included 49 questions on family demographics, oral health knowledge on the causes and consequences of caries, maternal and child nutrition and oral hygiene practices, and the mother’s perception of her child’s oral and overall health [20]. Surveys were conducted in Spanish by trained *promotoras* and health volunteers. Responses from participating mothers and data from exams were recorded on paper forms and entered into an Excel database.

Children’s height and weight, in light clothing and no shoes, were measured using a digital weighing scale and stadiometer by trained volunteers according to WHO anthropometric protocol [21]. Anthropometric data were entered into WHO AnthroPlus software (version 3.2.2) to generate standardized nutrition Z-scores, including weight-for-age Z-score (WAZ), height-for-age Z-score (HAZ), and weight-for-height Z-score (WHZ) [22]. Trained and licensed Salvadoran and U.S. dentists conducted a dental exam for each child, with headlamps and mirrors. No formal calibration procedures were undertaken. However, the dentists standardized their processes for examination by establishing an agreement on diagnostic criteria prior to data collection. During examinations, dentists were positioned side-by-side so that further comparison was possible and to reduce inter-examiner variability. If any unclear findings were present, examiners collaborated with one another to collectively decide upon the classification of a lesion. From each child’s dental exam, the presence of ECC was identified by the presence of one or more decayed (d), missing due to caries (m), or filled (f) primary teeth (t) [20]. For each child, the summed total number of decayed, missing, and filled teeth recorded was used as their “dmft score”. 

### 2.3. Statistical Analysis

Data were de-identified, cleaned, and analyzed on RStudio (version 1.0.143, RStudio PBC, Boston, MA, USA). The final study sample included baseline mother–child pair visits from 2006 to 2010; repeat visits over the years were excluded from the analysis to reduce intervention effects. Descriptive statistics were then generated to characterize the study population. Two different logistic regression models were built to examine nutrition and dental caries outcomes. 

The first model utilized ordinal logistic regression to identify risk factors for child undernutrition by examining ordinal weight-for-age Z-scores. Prior studies on undernutrition have typically used a binary outcome of “developed undernutrition” (WAZ, HAZ, or WHZ ≤ −2.0) and “did not develop undernutrition” (WAZ, HAZ, or WHZ > −2.0) [23,24]. In order to highlight more upstream factors that can influence a child’s risk for developing undernutrition, this analysis considers 3 outcomes: “developed undernutrition” (WAZ ≤ −2.0 Z-score), “at risk for undernutrition” (−2.0 < WAZ-score ≤ −1.0), and “did not develop undernutrition” (>−1.0 WAZ-score). Independent variables for the final multivariate model were selected based on a literature review that identified factors associated with undernutrition. Goodness of fit and the proportional odds assumption were assessed for the final model using the Lipsitz test and the Brant test, respectively [25,26]. 

The second model to identify risk factors for developing sECC utilized binary logistic regression; sECC was defined based on the total number of decayed, missing, or filled teeth (dmft) relative to the child’s age group: dmft ≥ 1 in children under age 3, dmft ≥ 4 at age 3, dmft ≥ 5 at age 4, and dmft ≥ 6 at age 5 [11]. Similar to the first model, covariates were selected based on factors associated with poor oral health found in existing literature. The Hosmer–Lemeshow method was used to test the model for goodness of fit, using a threshold of α < 0.05 to establish statistical significance for all analyses [25].

## 3. Results

The final cross-sectional analysis sample included baseline data from a total of 797 participating mother–child pairs from 2006 through 2010. Of these, approximately 21% were recruited in year 1, 26% were recruited in year 2, 16% were recruited in year 3, 15% were recruited in year 4, and 22% were recruited in year 5 of the program.

### 3.1. Demographic Characteristics

Table 1 shows the baseline demographics of the participating mothers and children. The mothers’ mean age was 29 years, and the children’s mean age was 2.9 years. Most mothers (58.2%) had some primary school level education, while 12.0% had no formal education and 29.8% had 8 years of schooling or more. The average household size consisted of 5–6 people. Slightly more than half (56.2%) of families had potable drinking water at home. Most families had electric lighting available at home (84.4%) and used some sort of wood fuel for cooking (86.8%).

### 3.2. Child Dietary Habits and Maternal Health Knowledge

Table 2 reports children’s caries-related dietary patterns and mothers’ oral health knowledge and perceptions. While 94.3% of children were ever breastfed (any attempt at breastfeeding, even if only for a short time), nearly one-half (42.6%) of children were also bottle-fed (mixed feeding). On average, children in this population were breastfed for 17.2 months; although the mean duration of bottle feeding was approximately 2 years (24.7 months), some children bottle fed until 4.5 years of age. One-third (33.3%) of the mothers reported that their children frequently drank from the baby bottle as they went to bed. Among bottle-fed children, the most common baby bottle contents were milk (80.2%), but approximately 3 in 4 children (73.0%) were given sugar-sweetened beverages (SSBs) in the baby bottle. SSBs reported in the baby bottle included sweetened *incaparina*: a commercial, fortified grain-based gruel powder (33.6%), juice with sugar (28.9%), lemonade with sugar (28.4%), sugar water (17.1%), soda (12.9%), and coffee with sugar (9.9%). Approximately half of children (54.4%) drank milk daily. Approximately one-third of children consumed candy/sweets (33.4%) and soda (35.4%) weekly, and more children ate candy/sweets at least once daily (22.2%) compared to soda (6.2%).

While a sizable portion of mothers reported knowing that not brushing teeth frequently (63.2%) and eating candy/sweets (29.6%) caused dental caries, far fewer mentioned that soda/juice (15.1%) or that using the baby bottle (12.9%) can cause caries. Most mothers noted that caries can negatively impact a child’s ability to eat (88.5%) and can cause mouth pain (53.2%); however, very few mothers reported knowing that caries could impact children’s quality of sleep (10.8%) or that caries were detrimental for their overall health (5.3%). Approximately one-quarter of mothers assessed their children’s oral health as “bad” for both oral health (25.8%) and overall health (26.2%).

### 3.3. Child Oral Health and Nutritional Status

Table 3 summarizes the nutritional and oral health status of participating children. While only 10.4% of the children experienced either moderate or severe undernutrition (WAZ ≤ −2), approximately one-fourth (24.1%) were at risk for undernutrition. The majority of children (65.4%) did not experience undernutrition, including 5.2% who were overweight or obese. Regarding oral health status, approximately half (47.4%) of children exhibited ECC. While 1 in 6 children experienced a dmft score between 1–3 (16.1%), 1 in 3 children (31.3%) had 4 or more decayed teeth, with a range up to 16 decayed teeth (i.e., decay in over ¾ of their total baby/primary teeth). One-third (31.0%) of the children experienced sECC. Almost all tooth decay (97.3%) was untreated, and approximately 1 in 5 children (19.3%) in this population experienced mouth pain.

### 3.4. Risk Factors for Child Underweight

After adjusting for covariates on socioeconomic demographics, the final multivariable model identified three factors was significantly associated with increased odds of being more underweight (i.e., either at risk of underweight or currently underweight). Children fed from baby bottles with SSBs had 2.26 times increased odds of being more underweight compared to those who were not fed SSBs in the baby bottle (OR: 2.26, 95% CI: 1.32–3.82). Higher numbers of decayed teeth (dmft scores) were also significantly associated with increased odds for being underweight; an additional dmft unit corresponded with 1.05 times increased odds (OR 1.05, 95% CI: 1.01–1.09). Living in a household that used wood-only fuel for cooking was also significantly associated with 1.60 times increased odds of being underweight compared to those from households that used gas only or a mix of wood and gas (OR 1.60, 95% CI: 1.14–2.23) (See Table 4 for detailed model results.). Full models were compared to nested models using the likelihood ratio test, and model goodness of fit was assessed using the Lipsitz test [25]. The proportional odds assumption was also satisfied for the final model (*p* = 0.39) [26]. Based on the results, the null hypothesis that ordinal logistic regression is a good fit for the sample population was not rejected (*p* = 0.19), and instead, it was concluded that the final model was a good fit for this population.

### 3.5. Risk Factors for Child Severe Caries Outcomes

Controlling for socioeconomic demographic covariates, this final multivariable model identified two factors significantly associated with increased odds of sECC. A unit increase in age was significantly associated with 1.57 increased odds of sECC (OR 1.57, 95% CI: 1.42–1.74). Children who were bottle-fed with SSBs in the baby bottle also had 1.53 times significantly higher odds of sECC compared to children that did not have SSBs in their baby bottle (OR 1.53, 95% CI: 1.07–2.18). Model fit was assessed using the Hosmer–Lemeshow test. Based on the results, the null hypothesis was unable to be rejected and there was no evidence that this model was a poor fit for the data (*p* = 0.11) [26] (See Table 5 for detailed model results).

ASAPROSAR *promotoras* also prepared a supplemental child case study illustrating the relationships among baby bottle use, sECC, and malnutrition (Appendix B).

## 4. Discussion

This study identifies factors for developing risk of undernutrition, moderate to severe undernutrition, and severe early childhood caries in a convenience sample of 797 children ages 6 months to 6 years in rural Salvadoran communities. Findings from this analysis highlight the interrelationship of children’s dietary consumption and their nutrition and oral health status. Specifically, consumption of SSBs in the baby bottle was significantly associated with undernutrition risk, severe to moderate undernutrition, and tooth decay. This study also found that tooth decay was significantly associated with both being at risk for undernutrition and with developing undernutrition. This study’s findings are consistent with existing literature, particularly regarding the consequences of the global nutrition transition in Latin America and other developing regions [27,28,29].

### 4.1. Sugar-Sweetened Beverages (SSBs), Bottle Feeding, and Adverse Health Outcomes

Past research has demonstrated the association between bottle-feeding, SSBs, and sweet snacks with ECC, which was formerly called “baby bottle tooth decay” [30,31,32]. Other studies have also identified a relationship between ECC and undernutrition, particularly in low- and middle-income countries [7], as well as associations of ECC with obesity, typically in higher-income populations [33]. In this study, sugar-sweetened beverages for bottle-feeding was significantly associated with undernutrition outcomes and sECC. While most children were breastfed, nearly half of the children were also bottle-fed, the majority were fed sugar-sweetened beverages in the baby bottle, half of children were given the bottle in bed, and some children were given the baby bottle until nearly age 5—all of which have previously been shown to contribute to sECC [27]. Though mothers were largely familiar with common causes of dental caries, most were unaware of the relationship between caries and sugar-sweetened beverages, especially when used for prolonged periods in the baby bottle. This study extends the current dialogue on the adverse nutritional consequences of sugar-sweetened beverages by identifying consumption in the baby bottle as a risk factor for developing undernutrition (lower WAZ scores), whereas most studies have cited the risk for obesity [34]. This study’s findings also highlight the increased importance of healthy bottle-feeding practices, as women globally transition to leaving the home for work and switching to bottle-feeding [35]. Frequent consumption of sugary drinks might contribute to undernutrition through suppressing children’s appetite for nutritious foods [36]. There are multiple possible physiological pathways: undernutrition can arise from sECC causing chronic infection/inflammation and mouth pain that can interfere with appetite and chewing ability, and chronic infection/inflammation can cause dysregulation of hepcidin and the hypothalamic–pituitary–adrenal hormonal axis.

### 4.2. Association between Early Childhood Caries (ECC) and Undernutrition

This study demonstrates an important relationship between children’s oral health and nutrition status. Half of the children were found to have ECC, one-third had severe ECC, and severe ECC was a significant risk factor for developing risk for undernutrition as well as moderate to severe undernutrition (OR 1.05). These findings show that, for each additional decayed tooth, a child has a 5% increased risk of undernutrition. While the mean number of decayed teeth (dmft) in this study was approximately 3 teeth, 31.4% of children had higher numbers of decayed teeth, up to 16 decayed teeth out of a total 20 baby/primary teeth—a sizable proportion of children with high numbers of decayed teeth who are at substantial additional risk for undernutrition.

### 4.3. Demographics and Socioeconomic Determinants

In this study, household use of wood fuel for cooking was a key risk factor for developing undernutrition in rural El Salvador. Prior studies have found that cooking with wood fuel is indicative of living in lower-resource settings or in low-GDP countries [37,38]. This finding underscores the need for health interventions to improve the social determinants of health and to reduce economic and health disparities [39]. Other demographic characteristics such as mother’s education and family size were not significantly associated with undernutrition or sECC. Our finding that mothers’ oral health knowledge was not significantly associated with children’s nutrition or oral health outcomes may be indicative of structural health barriers—such as, but not limited to, the lack of affordable nutritious foods and limited access to healthcare in rural areas—to ensuring good nutrition and oral health in rural El Salvador [18].

### 4.4. Recommendations

The shared set of risk factors between the two models in this study—for undernutrition and for ECC—also supports advocacy for more coordinated oral health and maternal–child health and nutrition programs, starting from pregnancy and birth onward. Maternal–child health professionals and *promotoras* should continue to promote breastfeeding, with clear guidelines for bottle-feeding when needed (including feeding babies only breastmilk or formula (not sugar-sweetened liquids) in the baby bottle, not putting the baby to bed with the bottle, and stopping the bottle after 12 months of age) [11]. This recommendation builds upon the Pan American Health Organization’s (PAHO) existing strategies for promoting healthy breastfeeding practices as part of their childhood obesity prevention efforts [40]. Maternal child–health interventions should also incorporate promotion of daily child toothbrushing with fluoride toothpaste starting at eruption of the first teeth around 6 months and that parents/caregivers assist children in brushing thoroughly until ages 6–8 [11,41]. In addition, maternal–child health programs should incorporate dental screening, oral health education, preventive fluoride varnish treatments, and referral to dental treatment as needed at well-child visits starting in infancy onward, as promoted by PAHO’s program Salud Oral Factores de Riesgo (SOFAR) [41,42].

Since 2008, the SOFAR initiative has trained cross-disciplinary health professionals—physicians, nutritionists, dentists, and social workers (among others)—to integrate oral health prevention with strategies for general health promotion by emphasizing evidence of shared risk factors. SOFAR advocates for coordinated, early interventions for pediatric populations through incorporating preventive oral health strategies (applying fluoride varnish and promoting oral health practices to parents) at regular primary care appointments for children [42]. Recent research has shown effectiveness in lowering pediatric oral disease through implementing integrated, interprofessional approaches [43].

Beyond further integration of maternal–child health and nutrition with oral health interventions, these findings also support a socioecological approach for developing interventions at multiple levels—in the home, community, national, and international landscapes—that address the socioeconomic and environmental factors influencing development of child undernutrition and sECC in rural El Salvador [44]. Past literature has indicated that the food environment at and surrounding schools is especially important for protecting children’s health [45]. Currently, all Latin American countries have agreed to a set of measures promoted by WHO and PAHO to prevent child obesity, including improving school food environments by strictly regulating labeling and advertising of processed snacks and beverages [40]. In June 2017, the Ministry of Education in El Salvador created a set of nutritional criteria to reduce children’s access to unhealthy beverages and foods high in sugar, fat, and salt in schools [46]. However, better enforcement of the policy is needed. Prior research has noted that effective implementation of school-based obesity-prevention programs suffers when there is limited integration of the children’s families and school vendors in the program’s efforts [47]. Additional policies are still needed to limit children’s access to unhealthy foods and beverages sold by vendors near the school grounds [18,45]. This can be supported by encouraging community access to fresh produce and by drafting additional restrictions for the marketing of fast foods and SSBs—a driver of the nutrition transition in Latin America [18,45,48,49]. The feasibility of public health interventions in El Salvador is also limited by the presence of context-specific challenges, including gang violence threatening implementation of sustainable development, high vulnerability to natural disasters severely impacting infrastructure and agriculture, and limited integration of national health systems [50]. Further research is also needed to assess the impact of community health programs on underserved communities and to foster a community-based participatory research model in which local stakeholders are engaged in the design and implementation of effective health interventions [51].

### 4.5. Study Limitations and Strengths 

The limitations of this study include convenience sampling, which limits the generalizability of findings. The cross-sectional study design is unable to establish causal relationships between risk factors and outcomes. Survey responses may be affected by recall and respondent biases. Due to time limitations in the mother interviews, the dietary consumption questions only included key cariogenic foods and drinks rather than a more comprehensive 24-h diet recall. Undernutrition outcomes could also have been more accurately modeled had the survey captured details on consumption of healthy foods, child infectious disease history, maternal malnutrition history and practices, and micronutrient deficiency [52]. As the study found wood-fuel cooking as a significant sociodemographic risk factor for developing undernutrition, the results interpret wood-fuel cooking as a proxy for low economic status. This interpretation is potentially limited by the proportion of families using wood fuel and differs from more conventional methods of using household assets to estimate income levels, which may have more accurately controlled for socioeconomic confounders in this analysis.

The study strengths are that the physical examination of children’s nutritional and oral health status in combination with the survey, including demographics, diet, practices, and perceptions, facilitated analysis of many associated risk factors for child nutrition and oral health outcomes. This study also builds upon existing literature by applying the ordinal logistic model to identify key risk factors for populations at risk for developing undernutrition. These findings can better guide early childhood health interventions to help mitigate risk for undernutrition and to prevent development of moderate to severe undernutrition. Longitudinal studies with a cohort representative of the population and additional research on the effectiveness of interventions to reduce consumption of SSBs in the baby bottle for the prevention of child undernutrition and sECC could build on these findings.

## 5. Conclusions

This study demonstrated the adverse impacts of the global nutrition transition on the nutritional status of a sample of young children in rural El Salvador. High rates of feeding young children sugar-sweetened beverages in the baby bottle were associated with increased risk for developing undernutrition and early childhood caries, and early childhood caries was further associated as a risk factor for undernutrition. To improve child nutrition, maternal–child health and nutrition programs should work with oral health programs to discourage feeding children sugar-sweetened beverages in the baby bottle, should advocate for policies to limit the marketing of sugar-sweetened beverages to children and families, and should ensure oral health screening and care for young children at well-child visits from infancy onward. Future research is needed to design more effective and sustainable community health programs to mitigate the consequences of the global nutrition transition.

## Figures and Tables

**Table 1 ijerph-18-00260-t001:** Mother–child participant demographics.

Participant Demographics (*n* Total Respondents)	*n* (%) or Mean ± s.d.
Mother and Family Characteristics	
Mother’s Age (years) (*n* = 591)	29.0 ± 8.9
Formal Education (mean) (*n* = 574)	5.6 ± 3.8
Formal Education (years) (*n* = 574)	
0	12.0%
1–3	19.3%
4–7	38.9%
8+	29.8%
Family Size (mean) (*n* = 582)	5.5 ± 2.4
Family Size (*n*) (*n* = 582)	
2–4	18.2%
5–7	64.4%
8+	17.4%
Potable Water at Home (*n* = 589)	56.2%
Electric Lighting at Home (*n* = 590)	84.4%
Cooking Fuel Source (*n* = 590)	
Wood Only	27.9%
Gas Only	13.2%
Both Wood and Gas	58.9%
Caregivers Receiving Prenatal Care (*n* = 797)	91.7%
Children with Completed Vaccinations (*n* = 797)	94.8%
Children who Visited Dentist At least Once (*n* = 796)	16.6%
Child Characteristics (*n* = 797)	
Age (mean)	2.9 ± 1.6
Age (years)	
(0.5–1]	12.3%
(1–2]	22.5%
(2–3]	20.1%
(3–4]	13.0%
(4–5]	14.9%
(5–6]	17.1%
Sex	
Male	51.1%
Female	48.9%

**Table 2 ijerph-18-00260-t002:** Children’s caries-related nutritional practice responses and mothers’ health knowledge and perceptions.

Nutrition/Oral Health Metric (*n* Total Respondents)	*n* (%) or Mean ± s.d.
Child Nutrition Practices (*n* = 793)	
Ever breastfed	94.3%
Mean breastfeeding duration (months)	17.2 ± 8.6
Range breastfeeding duration (months)	1–52
Ever bottle-fed	45.8%
Mixed bottle-fed and breastfed	42.6%
Mean bottle-feeding duration (months)	24.7 ± 12.7
Range bottle-feeding duration (months)	1–54
Bottle for sleeping ^1^	33.3%
Baby bottle contents ^1,2^	
SSBs ^3^ in the bottle	73.0%
Milk	80.2%
Incaparina ^4^	33.6%
Juice with sugar	28.9%
Lemonade	28.4%
Sugar water	17.1%
Soda	12.9%
Baby milk formula	9.9%
Coffee with sugar	9.9%
Children’s Frequency of Consumption ^1^	
Milk (*n* = 777)	
Infrequently (never or rarely)	30.4%
Weekly	15.1%
Daily	54.4%
Soda/juice (*n* = 799)	
Infrequently (never or rarely)	58.4%
Weekly	35.4%
Daily	6.2%
Candy/sweets (*n* = 784)	
Infrequently (never or rarely)	44.4%
Weekly	33.4%
Daily	22.2%
Maternal Oral Health Knowledge ^1^	
Causes of caries (*n* = 591)	
Not brushing frequently	63.2%
Candy/sweets	29.6%
Soda/juice	15.1%
Baby bottle	12.9%
Consequences of caries (*n* = 590)	
Trouble eating	88.5%
Mouth pain	53.2%
Rotting teeth	39.8%
Problems sleeping	10.8%
Poor overall health	5.3%
Maternal Perception of Children’s Health	
Mother’s opinion of their child’s teeth (*n* = 771)	
Excellent	38.9%
Okay	35.3%
Bad	25.8%
Mother’s opinion of their child’s overall health (*n* = 784)	
Excellent	36.8%
Okay	36.9%
Bad	26.2%

^1^ The open-ended question format allowed for multiple responses, and percentages reported indicate the percent (%) of mothers responding, not the % of responses; ^2^ Contents of the baby bottle were calculated by using only the 363 participants who bottle-fed their children in the denominator; ^3^ SSB: sugar-sweetened beverage; ^4^ Incaparina is a commercial, fortified grain-based gruel powder.

**Table 3 ijerph-18-00260-t003:** Children’s nutritional and oral health status (*n* = 797).

Descriptive Variable	*n* (%) or Mean ± s.d.
Nutritional Status	
Moderate or severe undernutrition (WAZ ≤ −2.0) ^1^	10.4%
At risk for developing undernutrition (WAZ ≤ −1.0) ^1^	24.1%
Did not develop undernutrition (WAZ > −1.0) ^1^	65.4%
Overweight or obese (WAZ > 2.0) ^1^	5.2%
Mean WAZ score ^1^	−0.6 ± 1.2
Mean HAZ score ^2^	−1.3 ± 1.2
Mean WHZ score ^3^	−0.1 ± 1.3
Oral Health Status	
Early childhood caries (ECC)	47.4%
Severe early childhood caries (sECC)	31.0%
Mean number of decayed teeth (dmft) ^4^	2.8 ± 3.9
Range number of decayed teeth	16
dmft 1–3	16.1%
dmft 4–6	13.9%
dmft 7–9	8.4%
dmft 10+	9.0%
Untreated tooth decay	97.3%
Reported mouth pain	19.3%

^1^ WAZ: weight-for-age z-score; ^2^ HAZ: height-for-age z-score; ^3^ WHZ: weight-for-height z-score; ^4^ dmft: decayed, missing, or filled teeth.

**Table 4 ijerph-18-00260-t004:** Risk factors for child underweight (low WAZ) ^1^ outcome (*n* = 563).

Co-Variates	Β ^2^	SE ^3^	OR ^4^	95% CI ^5^	*p*-Value
Intercept ^a^: Did not develop undernutrition | At risk for undernutrition	0.87	0.30	-	-	**0.003**
Intercept ^b^: At risk for undernutrition | Developed undernutrition	2.41	0.32	-	-	**0.000**
SSB ^6^ in baby bottle Reference: No SSB used SSB Used	0.81	0.32	2.26	1.32–3.82	**0.011**
Number of dmft ^7^	0.05	0.02	1.05	1.01–1.09	**0.023**
Cooking fuel source Reference: Both gas and wood Gas only Wood only	0.25 0.46	0.26 0.20	1.28 1.60	0.82–1.97 1.14–2.23	0.349 **0.022**
Child sex Reference: Female Male	0.13	0.18	1.13	0.85–1.52	0.479
Maternal education level Reference: No formal education 1–7 years 8+ years	−0.22 −0.13	0.28 0.33	0.81 0.88	0.52–1.28 0.51–1.52	0.442 0.700

Brant test for proportional odds assumption: Chi-square = 7.43, df = 7, and *p*-value = 0.39. Lipsitz test of goodness-of-fit of the overall model: Likelihood Ratio (LR) statistic = 12.49, df = 9, and *p*-value = 0.19. ^1^ WAZ: weight-for-age z-score; ^2^ B: Coefficient Value; ^3^ SE: Standard Error; ^4^ OR: Odds Ratio; ^5^ CI: Confidence Interval; ^6^ SSB: sugar-sweetened beverage; ^7^ dmft: decayed, missing, or filled teeth. Note: bold values are significant at *p* < 0.05. ^a^ Brant test for proportional odds assumption: Chi-square = 7.43, df = 7, and *p*-value = 0.39. ^b^ Lipsitz test of goodness-of-fit of the overall model: Likelihood Ratio (LR) statistic = 12.49, df = 9, and *p*-value = 0.19.

**Table 5 ijerph-18-00260-t005:** Risk factors for severe early childhood caries (sECC) outcome (*n* = 572).

Co-Variates	Β ^1^	SE ^2^	OR ^3^	95% CI ^4^	*p*-Value
Intercept: No sECC | sECC	−3.09	0.52	-	-	**0.000**
SSB ^5^ in baby bottle Reference: No SSB used SSB Used	0.43	0.21	1.53	1.07–2.18	**0.047**
Child age (continuous)	0.45	0.06	1.57	1.42–1.74	**0.000**
Maternal education level Reference: No formal education 1–7 years 8+ years	0.34 0.11	0.34 0.40	1.41 1.11	0.81–2.51 0.58–2.18	0.312 0.783
Cooking fuel source Reference: Both gas and wood Gas only Wood only	−0.34 0.02	0.31 0.26	0.71 1.02	0.42–1.16 0.67–1.56	0.262 0.933
Child sex Reference: Female Male	−0.34	0.20	0.71	0.51–0.99	0.091
In-home electric lighting Reference: Not available Available	0.52	0.33	1.68	0.99–2.91	0.114
Running potable water Reference: Not available Available	0.26	0.22	1.30	0.92–1.86	0.219

Hosmer–Lemeshow goodness-of-fit of the overall model: Chi-square = 14.47, df = 9, and *p*-value = 0.11; ^1^ B: Coefficient Value; ^2^ SE: Standard Error; ^3^ OR: Odds Ratio; ^4^ CI: Confidence Interval; ^5^ SSB: sugar-sweetened beverage. Note: bold values are significant at *p* < 0.05.

## Data Availability

The data presented in this study are available upon reasonable request to the corresponding author. The data are not publicly available due to privacy concerns for participants.

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
