# Peer review of "Sugary Liquids in the Baby Bottle: Risk for Child Undernutrition and Severe Tooth Decay in Rural El Salvador"

_ijerph, 2020, doi:10.3390/ijerph18010260_

Round 1

Reviewer 1 Report

First, I would like to thank you for the opportunity to review this manuscript and congratulate the authors for the article.

This study aims to identify dietary, oral health, and socio-demographic risk factors for child undernutrition and severe early childhood caries (sECC) for rural communities in El Salvador. The authors observed high rates of feeding young children sugar sweetened beverages in the baby bottle were associated with increased risk for both worse undernutrition outcomes and early childhood caries, and early childhood caries was further associated as a risk factor for worse nutrition.

I have some suggestions:

[1] Abstract: Please state the objective of the study in the abstract.

[2] Abstract: Please state which software were used for statistical analysis.

[3] Abstract: Please provide the p value of each association.

[4] Introduction: Well-written. I only suggest that the authors state the null hypothesis of the study.

[5] Methods: Please describe with more details the inclusion and exclusion criteria.

[6] Methods: Please provide the acceptable inter-examiner variability considered in the study.

[7] Results: How many participants were contacted? Were all of them included? If not, which were the reasons for not inclusion/exclusion? Please state.

Author Response

Reviewer 1:
1. First, I would like to thank you for the opportunity to review this manuscript and congratulate the authors for the article. This study aims to identify dietary, oral health, and socio-demographic risk factors for child undernutrition and
severe early childhood caries (sECC) for rural communities in El Salvador. The authors observed high rates of feeding young children sugar sweetened beverages in the baby bottle were associated with increased risk for both worse undernutrition outcomes and early childhood caries, and early childhood caries was further associated as a risk factor for worse nutrition. I have some suggestions:
Response: We appreciate the reviewer’s feedback and are committed to making the necessary revisions.

2. Abstract: Please state the objective of the study in the abstract.
Response: We have added the following sentence to the abstract in order to clarify the study’s objective:
• Lines 21-24: “This study aims to identify dietary, oral health, and socio-demographic risk factors for child undernutrition and severe early childhood caries (sECC) among a convenience sample of 797 caregiver-child pairs from rural Salvadoran communities.”

3. Abstract: Please state which software were used for statistical analysis.
Response: We have added the following phrase within the abstract to specify the software used:
• Lines 25-26: “Multivariable regression analyses were performed using RStudio (version 1.0.143).”

4. Abstract: Please provide the p value of each association.
Response: We have included p-values for the significant risk factors mentioned in the abstract. Please find the additions in the following locations:
BERKELEY • DAVIS • IRVINE • LOS ANGELES • MERCED • RIVERSIDE • SAN DIEGO • SAN FRANCISCO SANTA BARBARA • SANTA CRUZ

• Lines 26-27: “Caregiver use of SSB’s in the baby bottle… for child undernutrition (P = .011)”
• Line 28: “sECC (P = .047)”
• Line 28: “Early childhood caries (P = .023)”

5. Introduction: Well-written. I only suggest that the authors state the null hypothesis of the study.
Response: Thank you for your comment. Since the models in our study are aimed at identifying potential covariates that may be associated with the outcomes (undernutrition and severe early childhood caries), it may not be
appropriate to state the null hypothesis prior to conducting the analysis. Instead, our null hypothesis as it stands in the paper states that the models built to conduct these analyses are not a good fit for the data, which would limit the
inference we might be able to draw from it.

6. Methods: Please describe with more details the inclusion and exclusion criteria.
Response: We have added the following phrase regarding exclusion criteria to the “Study population” subsection within the Methods:
• Lines 86-87: “Children below the age of 6 months, and at or above the age of 6 years were excluded from the study population, but were given nutrition and oral health education.”

7. Methods: Please provide the acceptable inter-examiner variability considered in the study.
Response: We have added the following clarification on inter-examiner variability to the Methods within the “Data Collection” subsection:
• Lines 105-110: “No formal calibration procedures were undertaken. However, the dentists standardized their processes for examination by establishing agreement on diagnostic criteria prior to data collection. During examinations, dentists were positioned side-by-side so that further comparison was possible and to reduce interexaminer variability. If any unclear findings were present, examiners collaborated with one another to collectively decide upon the classification of a lesion.”

8. Results: How many participants were contacted? Were all of them included? If not, which were the reasons for not inclusion/exclusion? Please state.
Response: Our study sampling was primarily intended to be a convenience sample of participants who were able to attend the health camp, from the initial dissemination of information to the community by the non-profit partner
(ASAPROSAR). Because of this snowball-like sampling, we are unable to quantify the number of people that were initially contacted. However, as described above, we excluded any children below the age of 6 months or at/above the age of 6 years.

Reviewer 2 Report

Overall, this is a very interesting study addressing the socioeconomic factors associated with nutritional status and oral health in children from rural region of El Salvador. The methods and results are very well presented and easy to follow. I have some minor comments that could improve the clarity of this manuscript.

  1. The terms ‘undernutrition’ and ‘malnutrition’ are used interchangeably. In this study weight for age Z score (WAZ) were used as a proxy measure of nutritional status (not undernourished, at risk for undernourished, moderately undernourished and overweight and obese). The authors need to use clarify the use of proxy measure and be consistent with the terminology.
  2. First, authors have provided a great deal of recommendations on improving oral health and nutritional status of children in rural areas of El Salvador. However, some of these recommendations are not suitable for population in rural areas with limited resources and knowledge. For example, the recommendations of screening children during the well visit needs more information on whether these populations have access to health care system for regular check-up. Second, the authors need to address the complexity of the issue and highlight the barriers in addressing undernutrition and oral health in such populations. For example, the recommendations of reducing marketing of SSB does not seem to be suitable here. The behavior of adding sugary drinks to the bottle may be a result of more complex issues such as lack of nutritious foods, affordability of nutritious foods, parental lack of knowledge etc. Third, the authors provide recommendations for reducing obesity in children by collaborating with school system, which is a great but not suitable to the scope of this study as it addresses undernutrition related to the parental/caregivers’ behaviors (of adding sugary drinks to the bottle) and hence the recommendations need to be focused on the topic of the manuscript.

Here are some specific minor comments

Methods:

Line 102 and 103: How was dmft score created? Did each condition (decayed, missing and filled) have a specific score based on the condition? Given the importance of the variable, some information on scoring needs to be provided.

Authors need to explain what comprises of nutritional status. It seems the nutritional status was assessed using the anthropometric indices. This needs to be explained in the introduction as well as methods (line 99)

Line 124: Were children age 6 years included in the study? If so, how did they fit into the dmft category for logistic regression?

Is it possible to provide number of children who were both breastfed and bottle fed?

Discussion:

Line 282-292: While this paragraph point to important efforts to prevent childhood obesity in Latin American countries, it does not seem to be relevant to the scope this study or its findings. It would be advisable to elaborate on efforts of PAHO and their target population and its effectiveness (if measured). Furthermore, authors have provided a great deal of preventive strategies in lines 261-272. Please provide more information on feasibility of implementing these strategies in El Salvador or other the Latin American countries with similar socio-economic background.

The term worse nutrition/worse undernutrition is not an appropriate term and has been used several times in the manuscript. Please use appropriate terms. For example, ‘ECC was further associated with undernutrition’.

Line 318: Authors are recommending to advocate for policies to limit marketing of sugar sweetened beverages, which could be a potential problem leading to increased consumption of sugar sweetened beverages (SSB) in children in many parts of the world. However, authors did not provide background information on marketing SSB as a potential problem in increased bottle feeding of SSB in El-Salvador, particularly in rural areas such as Santa Ana region. Probably, marketing may not be the issue here. It could be lack of nutritious options, lack of awareness among mothers (which the study found) etc. which may lead the mothers to make the choice of bottle feeding SSB to children, especially at an older age of 5. If marketing is indeed an issue, please provide background information to strengthen the point. Furthermore, the authors’ suggestion to ensure oral health screening and care for young children at well-child visits from infancy onward also warrants some background, particularly in rural El-Salvador. Please provide some background on health care access of mothers and children and how these strategies can be incorporated given that access.

Author Response

Reviewer 2:
1. Overall, this is a very interesting study addressing the socioeconomic factors associated with nutritional status and oral health in children from rural region of El Salvador. The methods and results are very well presented and easy to follow. I have some minor comments that could improve the clarity of this manuscript.
Response: Thank you for your feedback. We appreciate the reviewer’s interest in the study and are committed to making the required revisions.

2. The terms ‘undernutrition’ and ‘malnutrition’ are used interchangeably. In this study weight for age Z score (WAZ) were used as a proxy measure of nutritional status (not undernourished, at risk for undernourished, moderately undernourished and overweight and obese). The authors need to use clarify the use of proxy measure and be consistent with the terminology.
Response: We agree that the text’s current use of terminology (undernutrition vs. malnutrition) should be better clarified. For this study, we have defined “malnutrition” to describe the nutritional state where a deficiency or an excess of nutrients cause adverse effects on an individual’s clinical health outcomes. We intended to reference undernutrition as the subtype of malnutrition where there is a deficiency of the above described energy and/or nutrients. In order to better clarify this, we have made the following changes to the text:
• Lines 42-43: We have added the definition of “malnutrition: the deficiency, excess, or imbalance in an individual’s consumption of energy and nutrients [3]” to the Introduction
• Lines 46-47: We have also added this clarifying phrase: “malnutrition—which includes both overweight/obesity and undernutrition…” to the Introduction.
• Global change throughout the text: In order to maintain consistent use of terminology, we have also replaced any reference to “undernourishment” with “undernutrition.”
Response: Regarding Reviewer 2’s point on “proxy measure,” we would like to emphasize that weight-for-age Zscore (WAZ) is a standard anthropometric measure, and not a proxy, of acute and chronic malnutrition. Therefore,
we have used WAZ scores to assign nutrition status for participating children. We have specified the proportion of “overweight” and “obese” children in Table 3 for interested readers; however, when developing the WAZ model on undernutrition, both overweight and obese children were captured in the “Did not develop undernutrition” category.

3. First, authors have provided a great deal of recommendations on improving oral health and nutritional status of children in rural areas of El Salvador. However, some of these recommendations are not suitable for population in rural areas with limited resources and knowledge. For example, the recommendations of screening children during the well visit needs more information on whether these populations have access to health care system for regular checkup.
Response: We agree that information contextualizing access to the health system is needed to improve the relevance of our findings. As part of our survey, our team collected information on caregivers’ utilization of well-visits (for prenatal care and children’s vaccinations) and on children’s dental visit history. Results showed that while a majority of caregivers received prenatal care (91.7%) and a majority of children had completed their vaccinations (94.8%), only 1 in 5 children had ever been to the dentist (16.6%). As such, our findings suggest that preventive oral health care be incorporated into the existing maternal-child health system for well-visits, rather than build separate infrastructure for dental care We have incorporated these descriptive statistics into Table 1. These findings further support the work done by PAHO through their SOFAR initiative, as explained in the
Discussion. Since 2008, the SOFAR initiative has trained cross-disciplinary health professionals—physicians, nutritionists, dentists, social workers (among others)—to integrate oral health prevention with strategies for general health promotion by emphasizing evidence of shared risk factors. SOFAR advocates for coordinated, early interventions for pediatric populations through incorporating preventive oral health strategies (applying fluoride varnish and promoting oral health practices to parents) at regular primary care appointments for children. Additionally, since the beginning of this program, ASAPROSAR’s (our community nonprofit partner) promotoras were trained to incorporate oral health into their usual maternal-child health and nutrition promotion. They did this very effectively, as illustrated in the case study (Appendix A).
• Table 1 (Line 149): “Caregivers Receiving Prenatal Care: 91.7%; Children with Completed Vaccinations: 94.8%; Children who Visited Dentist At least Once: 16.6%”

4. Second, the authors need to address the complexity of the issue and highlight the barriers in addressing undernutrition and oral health in such populations. For example, the recommendations of reducing marketing of SSB does not seem
to be suitable here. The behavior of adding sugary drinks to the bottle may be a result of more complex issues such as lack of nutritious foods, affordability of nutritious foods, parental lack of knowledge etc.
Response: We agree that there are a multitude of factors impacting the caregiver behavior of adding SSB’s to the baby bottle, and have amended the Discussion to discuss structural health barriers that exacerbate the behavior of adding sugary drinks to the bottle, along with updated reference literature. While we acknowledge that there might be more upstream, supply-side factors that change the local food environment and influence this practice, prior studies have discussed the additional detriments of changing food demands in Latin America, attributable to an increased exposure in targeting marketing and advertisements. Given the context of the study, and the relative feasibility of policy interventions to address this, compared to some of the community and system-related factors, we feel that SSB marketing is an important discussion point. Our research team’s prior study (An interpretative study on food, snack, and beverage advertisements in rural and urban El Salvador – Amanzadeh et. al) found that pervasive SSB advertisements contributed to the increased consumption of SSB’s in rural El Salvador. In a second study with a similar study population of rural Salvadoran
caregivers from neighboring Santa Ana communities, our team also found widespread sales of SSB’s and targeted SSB marketing to be prominent barriers experienced by caregivers trying to encourage healthy beverages choices for
their children (A Qualitative Study of Child Nutrition and Oral Health in El Salvador – Achalu et. al). However, similar to the Reviewer’s comments, the second study above also noted other significant barriers—including the cost of nutritious foods. We have now updated the Discussion to better reflect the impact of these additional factors:
• Lines 303-307: In line with the socio-ecological model, we have also incorporated the sentence “Beyond further integration of maternal-child health and nutrition with oral health interventions, these findings also support a
socio-ecological approach for developing interventions at multiple levels—in the home, community, national, and international landscape—that address the socioeconomic and environmental factors influencing development of child undernutrition and sECC in rural El Salvador [44]” to better address that several factors contribute to the consumption of SSB’s and that interventions at multiple levels are needed—from child to family, community, and national/global levels. Our discussion section captures some of the feasible recommendations but acknowledges there are several, complex variables influencing poor child nutrition and oral health beyond what we have discussed
• Line 276-277: We have also added the phrase “structural health barriers—such as lack of affordable nutritious foods and limited access to healthcare in rural areas” to acknowledge alternative factors influencing nutrition and oral health outcomes
• Lines 319-320: We have added the clarifying phrase “marketing of fast foods and SSB’s—a driver of the nutrition transition in Latin America [18, 45, 48, 49]” to incorporate new references supporting the case that SSB marketing is a relevant driver to consider

5. Third, the authors provide recommendations for reducing obesity in children by collaborating with school system, which is a great but not suitable to the scope of this study as it addresses undernutrition related to the parental/caregivers’ behaviors (of adding sugary drinks to the bottle) and hence the recommendations need to be focused on the topic of the manuscript.
Response: Taking into account earlier comments from Reviewer 2, we have elaborated on interventions that may influence consumption of SSB’s, in addition to the individual-level behavioral practices. Part of this practice is attributable to an increased demand for SSB’s due to the surge in targeted advertisement and marketing (Nutrition Status of Children in Latin America – Corvalán et. al). However, there still remain supply-side factors in Latin America that impact the local food environments to facilitate healthier food choices. We expand the scope of our study recommendations to discuss both and take into account some of these issues. Though our findings isolate sugar-sweetened beverage (SSB) use in the baby bottle as a key risk factor for child undernutrition, past literature has identified SSB consumption as a significant risk factor for developing child obesity
as well. As outlined in the Introduction, El Salvador’s increasing obesity burden remains a pertinent problem along with its persistence of undernutrition. Reduction of SSB consumption through the baby bottle is the immediate
implication of this study’s results, but given the broader child nutrition landscape of El Salvador, we feel that collaborating with school systems to encourage healthier food/beverage consumption choices among older children (who are not still being bottle-fed) is still relevant to the study’s scope. Most of the 3-5 year-old children in our study (accounting for roughly half of our study population) were in preschool or kindergarten. Some of the schools sold SSB/junk food on school grounds (counter to policy guidance), and SSB/junk food vendors had set up their shops immediately outside the school as well. In a separate study, it was noted
most parents also gave their children money to purchase snacks at school (A Qualitative Study of Child Nutrition and Oral Health in El Salvador – Achalu et. al). This underscores the widespread exposure of children to sugary drinks—particularly at school—and the need for enforcing policies to ensure that the school environment provides healthy nutrition to support families in healthy feeding behaviors. We have made the following changes:
• Line 307-308: We have added the sentence “Past literature has indicated that the food environment at and surrounding schools is especially important for protecting children’s health [45]” to address why we are discussing school-based recommendations. We have also incorporated a new reference to support this claim.
• Line 316-317: We have cited additional studies to support the claim that better enforcement of school policies protecting children’s nutrition is needed, both in Latin America more broadly and specifically for rural El Salvador: “Additional policies are still needed to limit children’s access to unhealthy foods and beverages sold by vendors near the school grounds [18, 45].”

6. Here are some specific minor comments. Methods: Line 102 and 103: How was dmft score created? Did each condition (decayed, missing and filled) have a specific score based on the condition? Given the importance of the variable, some information on scoring needs to be provided.
Response: The dmft score used in the study was based on methodology outlined for the standard dental caries index, as defined by the World Health Organization in their report: “Oral health surveys: basic methods – 5th edition.” As mentioned in Lines 110-113, examiners performed dental exams on each child. For every child, each of their teeth was assigned a status: “healthy,” “decayed,” “missing,” or “filled.” After the data for each tooth was collected, a single dmft score was created for each child. This was done by summing the total number of teeth classified as “decayed,” “missing,” or “filled” for each child.
• Lines 112-113: We have added the clarifying phrase “sum total” to the sentence “For each child, the sum total of the number of decayed, missing, and filled teeth recorded was used as their “dmft score” to make the methodology behind the dmft score calculations clearer

7. Authors need to explain what comprises of nutritional status. It seems the nutritional status was assessed using the anthropometric indices. This needs to be explained in the introduction as well as methods (line 99).
Response: As recommended by the World Health Organization, we were able to define nutritional status by using a child’s height, weight, and/or age data. With this data, researchers can assess children’s nutritional status through standardized age- and sex-specific growth reference to calculate weight-for-age Z-scores (WAZ), height-for-age Zscores (HAZ), and weight-for-height Z-scores (WHZ). While we did calculate all three of these standardized measures (shown in Table 3), we have only used WAZ score when defining undernutrition status for our regression model. As the calculation of this standardized score is how the various nutrition outcomes (i.e. Developed undernutrition, At risk for undernutrition, and Did not develop undernutrition) were defined, we feel any explanation of their calculation is not appropriate for the Introduction. However, to better clarify this in the Methods, we have changed the following:
• Line 103-104: We have added the phrase “including weight-for-age Z-score (WAZ), height-for-age Z-score (HAZ), and weight-for-height Z-score (WHZ)” to specify the different standardized nutrition Z-scores assessed in the study
• Lines 122-123: We have added more specific definitions to each outcome by explicitly indicating that the binary definition of nutrition status in other studies utilizes the following cutoffs: “WAZ, HAZ, or WHZ ≤ -2.0” as “Developed undernutrition” and “WAZ, HAZ, or WHZ > -2.0” as “Did not develop undernutrition.”
• Lines 125-126: We have better clarified that for our analysis, we utilize the following three outcome definitions based only on WAZ score: “Developed undernutrition” (WAZ ≤ -2.0 Z-score), “At risk for undernutrition” (-2.0 <
WAZ-score ≤ -1.0), and “Did not develop undernutrition” (> -1.0 WAZ-score).

8. Line 124: Were children age 6 years included in the study? If so, how did they fit into the dmft category for logistic regression?
Response: Children who had turned 6 years old (or beyond) were not included in the study. We have better clarified this cutoff by making the following change in the Methods:
• Line 86-87: We have modified the exclusion criteria to read: “Children below the age of 6 months, and at or above the age of 6 years were excluded from the study population…”

9. Is it possible to provide number of children who were both breastfed and bottle fed?
Response: Of the 748 children breastfed, 319 of these children were also bottle-fed. We have incorporated this detail at the following locations:
• Lines 152-153: We have reported the proportion of children who were both bottle- and breastfed with the phrase “nearly one-half (42.6%) of children were also bottle-fed (mixed feeding).”
• Table 2 (Line 173): We have added the N (%) describing this population with “Mixed bottle-fed and breastfed: 42.6%”

10. Line 282-292: While this paragraph point to important efforts to prevent childhood obesity in Latin American countries, it does not seem to be relevant to the scope this study or its findings. It would be advisable to elaborate on efforts of PAHO and their target population and its effectiveness (if measured).
Response: Similar to our response to Reviewer 2’s earlier comment (#5), we feel that discussion of childhood obesity prevention strategies is still relevant to the scope of this paper. Research has identified that while child undernutrition
persists, childhood obesity is increasing in many low-to-income communities worldwide, including in El Salvador. As a result, though our results identify key risk factors for developing undernutrition, it is still important to consider the
impact these findings may have for malnutrition—which includes undernutrition and overweight/obesity— more broadly. The recommendations from our study—namely, promotion of healthy breastfeeding practices; incorporation of dental care into maternal-child well visits; and increased regulation of sugar-sweetened beverages/processed snacks at schools—are aligned with PAHO’s current efforts. To better emphasize this, we have incorporated additional detail highlighting this alignment with PAHO’s strategies as shown below. We will address the Reviewer’s
comment on feasibility/effectiveness in the next section (#11):
• Lines 286-288: We have added the sentence “This recommendation builds upon the Pan American Health Organization’s (PAHO) existing strategies for promoting healthy breastfeeding practices as part of their childhood obesity prevention efforts [40]” to highlight that recommendations on healthy breastfeeding promotion align with PAHO’s current work
• Lines 295-300: The following paragraph details efforts by PAHO in their SOFAR program to advocate for interdisciplinary collaboration between nutrition and oral health: “Since 2008, the SOFAR initiative has trained cross-disciplinary health professionals—physicians, nutritionists, dentists, social workers (among others)— to integrate oral health prevention with strategies for general health promotion by emphasizing evidence of shared risk factors. SOFAR advocates for coordinated, early interventions for pediatric populations through incorporating preventive oral health strategies (applying fluoride varnish and promoting oral health practices to
parents) at regular primary care appointments for children”
• Lines 308-311: The following sentence outlines PAHO’s efforts to regulate food/SSB sales in schools: “Currently, all Latin American countries have agreed to a set of measures promoted by WHO and PAHO to prevent child obesity, including improving school food environments by strictly regulating labeling and advertising of processed snacks and beverages [40].”

11. Furthermore, authors have provided a great deal of preventive strategies in lines 261-272. Please provide more information on feasibility of implementing these strategies in El Salvador or other the Latin American countries with
similar socio-economic background.
Response: We agree that additional acknowledgement of the feasibility challenges these preventive efforts can and do run into, especially in limited resource settings, can strengthen our discussion. As such, we have made the following
changes to the manuscript:
• Lines 314-316: We have cited an additional Mexico-based study to highlight potential challenges in implementing such school-based obesity prevention programs through discouraging SSB consumption: “Prior research has noted that effective implementation of school-based obesity-prevention programs suffers when there is limited integration of the children’s families and school vendors in the program’s efforts [47].”
• Lines 320-323: We have also added the following phrase to delineate other, El Salvador-specific challenges for implementing public health strategies: “The feasibility of public health interventions in El Salvador is also limited by the presence of context-specific challenges, including gang violence threatening implementation of sustainable development; high vulnerability to natural disasters severely impacting infrastructure and agriculture; and limited integration of national health systems [50].”

12. The term worse nutrition/worse undernutrition is not an appropriate term and has been used several times in the manuscript. Please use appropriate terms. For example, ‘ECC was further associated with undernutrition’.
Response: We have replaced the phrasing “worse nutrition” and “worse undernutrition” with “undernutrition” globally throughout the manuscript.

13. Line 318: Authors are recommending to advocate for policies to limit marketing of sugar sweetened beverages, which could be a potential problem leading to increased consumption of sugar sweetened beverages (SSB) in children in many parts of the world. However, authors did not provide background information on marketing SSB as a potential problem in increased bottle feeding of SSB in El-Salvador, particularly in rural areas such as Santa Ana region.
Probably, marketing may not be the issue here. It could be lack of nutritious options, lack of awareness among mothers (which the study found) etc. which may lead the mothers to make the choice of bottle feeding SSB to children, especially at an older age of 5. If marketing is indeed an issue, please provide background information to strengthen the point. Furthermore, the authors’ suggestion to ensure oral health screening and care for young children at well-child visits from infancy onward also warrants some background, particularly in rural El-Salvador. Please provide some background on health care access of mothers and children and how these strategies can be incorporated given that access.
Response: We agree that there are a number of additional factors influencing a caregiver’s decision to use SSB’s in the baby bottle. As mentioned in our earlier response to Reviewer 2 (please refer to point #4), we have acknowledged
the presence of these additional factors. We also want to emphasize that the recommendation to better regulate SSB marketing is just one piece of multi-factorial intervention to improve nutrition outcomes in El Salvador. While our
research focuses on the caregiver use of SSB’s in the baby bottle, a socio-ecological model calls for consideration of contributing factors across multiple levels (from child to family, community, nation/global levels) and corresponding
interventions needed at these various levels. This would include educational interventions for families but also policies implemented at the community, school, health program, and national levels. Prior research has found a surge
in targeted SSB/junk food marketing towards young children/ their families in El Salvador (An interpretative study on food, snack, and beverage advertisements in rural and urban El Salvador – Amanzadeh et. al). Research has also indicated that caregivers cited pressures from SSB marketing as a barrier to children’s nutrition and oral health in rural El Salvador (A Qualitative Study of Child Nutrition and Oral Health in El Salvador – Achalu et. al). We have incorporated these new references to better support the relevance of discussion on SSB marketing as shown below:
• Lines 303-308: We have added that several variables acting at different levels are influencing children’s nutrition and oral health in this population. As such, this calls for interventions implemented at various levels, as acknowledged by the following: “Beyond further integration of maternal-child health and nutrition with oral health interventions, these findings also support a socio-ecological approach for developing interventions at multiple levels—in the home, community, national, and international landscape—that address the socioeconomic and environmental factors influencing development of child undernutrition and sECC in rural El Salvador [44]. Past literature has indicated that the food environment at and surrounding schools is especially important for protecting children’s health [45].”
• Lines 318-319: We have added the phrase: “drafting additional restrictions for the marketing of fast foods and SSB’s—a driver of the nutrition transition in Latin America [18, 45, 48, 49],” supported by the above explained references.
Please refer to our earlier response to Reviewer 2’s comment (#3) for additional context on healthcare access for mothers and children, and how oral health checkpoints can be feasibly incorporated into these existing maternal child-health well visits.

Reviewer 3 Report

General comments –The paper in its current form lacks strength in its structure overall and it would be helpful if the authors could incorporate a more critical and appropriate arguments to justify the findings of the study. Some specific comments to consider are as follows:  

Specific comments –

Introduction:

  • The introduction provides very little evidence to make a strong case for the purpose of the study. A clear definition of undernutrition, malnutrition, ECC and sECC is missing. Also, some discussion is needed on how undernutrition is different from malnutrition; and ECC is different from sECC?
  • It is unclear if and why terms undernutrition and malnutrition are used interchangeably? Needs clarification
  • A paragraph to build the context of sugary liquids in the baby bottle leading up to the rationale of the study will be helpful for readers.

Methods:

  • A detailed explanation of what standardized processes were undertaken to establish inter-examiner reliability is needed?
  • How was ECC and sECC scored/ measured?
  • A detailed explanation/ rationale for the choice of variables (such as electric lighting etc.) included in the analysis is needed?

Results:

  • Include n for each variable in Table 1 and 2.
  • Explain what is meant by ‘ever breastfed’?
  • Page 4, line 156: ‘baby bottle (12.9%) caused caries’ seems unclear?

Discussion:

  • Findings need a more through discussion and must be adequately supported by the literature.
  • The section is currently unorganised and incoherent.
  • Several findings need clear discussion, supported with evidence and clarity, such as:
    • WHY Children fed from baby bottles with SSB’s had 2.26 times increased odds of being more underweight?
    • WHAT is meant by ‘Healthy bottle-feeding practices’?
    • Mothers’ oral health knowledge was not significantly associated with nutrition or oral health outcomes (OF ??) may be indicative of the structural health barriers (WHAT ARE THESE??) to ensuring good oral health in rural El Salvador.
  • Support with evidence ‘wood-fuel cooking as a proxy for low economic status’
  • No acknowledgement that the results could be potentially confounded by measures (give examples?) not captured in the dataset.
  • Paragraph on implications for future research and policy needs strengthening and discussion.

Author Response

Reviewer 3:
1. General comments – The paper in its current form lacks strength in its structure overall and it would be helpful if the authors could incorporate a more critical and appropriate arguments to justify the findings of the study. Some specific comments to consider are as follows:
Response: We thank the Reviewer for their feedback and are committed to adding supporting details to strengthen the relevance and presentation of our findings.

2. The introduction provides very little evidence to make a strong case for the purpose of the study. A clear definition of undernutrition, malnutrition, ECC and sECC is missing. Also, some discussion is needed on how undernutrition is different from malnutrition; and ECC is different from sECC?
Response: Thank you for your suggestions. We agree that the current use of terminology for undernutrition and malnutrition can be improved. We have added the appropriate clarifications to the Introduction, as shown below. In the Methods, we have detailed how both early childhood caries (Lines 110-113) and severe early childhood caries (Lines 132-134) are defined. Specifically, presence of early childhood caries is indicated when there is one or more decayed, missing due to caries, or filled primary tooth (dmft) present. Severe early childhood caries occurs when a certain threshold of dmft is present. Guidelines for classifying sECC for each age group is further delineated in the Methods (Lines 132-134).
• Lines 42-43: We have introduced the following definition of “malnutrition: the deficiency, excess, or imbalance in an individual’s consumption of energy and nutrients [3].”
• Lines 46-47: We have also specified that malnutrition consists both of the deficiency (undernutrition) and excess (overweight/obesity) of the above specified energy and nutrients by indicating the following: “malnutrition—which includes both overweight/obesity and undernutrition”
• Lines 112-113: We have added the clarifying phrase “sum total” to the sentence “For each child, the sum total of the number of decayed, missing, and filled teeth recorded was used as their “dmft score” to make the methodology behind the dmft score calculations clearer

3. Introduction: It is unclear if and why terms undernutrition and malnutrition are used interchangeably? Needs clarification
Response: We have incorporated the following clarifications to explain the difference in terminology. For this study, we have defined “malnutrition” to describe the nutritional state where a deficiency or an excess of nutrients cause
adverse effects on an individual’s clinical health outcomes. We intended to reference undernutrition as the subtype of malnutrition where there is a deficiency of the above described energy and/or nutrients. For this study, our weightfor-age Z-score model analyzes children’s undernutrition only. We have ensured consistent references to “undernutrition” and not “malnutrition” when discussing the results of our regression model throughout the text. In order to better clarify this, we have also made the following changes to the Introduction:
• Lines 42-43: We have introduced the following definition of “malnutrition: the deficiency, excess, or imbalance in an individual’s consumption of energy and nutrients [3].”
• Lines 46-47: We have also specified that malnutrition consists both of the deficiency (undernutrition) and excess (overweight/obesity) of the above specified energy and nutrients by indicating the following: “malnutrition—which includes both overweight/obesity and undernutrition”

4. Introduction: A paragraph to build the context of sugary liquids in the baby bottle leading up to the rationale of the study will be helpful for readers.
Response: We would like to emphasize that the original aim of the study was to conduct an exploratory analysis considering dietary, oral health, and sociodemographic risk factors associated with undernutrition and severe early
childhood caries. Since this analysis was exploratory, we did not begin the analysis stage with the intention of honing in on any specific risk factor (including use of sugar-sweetened beverages in the baby bottle). After building the models, we were then able to isolate SSB use in the baby bottle as a key risk factor. As such, we feel that discussion of sugary liquids in the baby bottle is better suited in its current position within the Discussion, rather than in the Introduction. However, in order to more clearly emphasize the exploratory nature of this analysis, we have made the following changes to the Introduction:
• Lines 70-72: We have updated the aim of the study to the following: “This study is an exploratory analysis aiming to identify dietary, oral health, and socio-demographic risk factors for child undernutrition and severe early childhood caries (sECC) for rural communities in El Salvador.”

5. Methods: A detailed explanation of what standardized processes were undertaken to establish inter-examiner reliability is needed?
Response: We have added the following clarification on inter-examiner variability to the Methods within the “Data Collection” subsection:
• Lines 105-110: “No formal calibration procedures were undertaken. However, the dentists standardized their processes for examination by establishing agreement on diagnostic criteria prior to data collection. During examinations, dentists were positioned side-by-side so that further comparison was possible and to reduce interexaminer variability. If any unclear findings were present, examiners collaborated with one another to collectively decide upon the classification of a lesion.”

6. Methods: How was ECC and sECC scored/ measured?
Response: In the Methods section, we have detailed the calculations used to assess both ECC and sECC (Lines 110-113; Lines 132-134). To confirm, presence of early childhood caries is indicated when one or more decayed (d), missing (m) due to caries, or filled (f) primary tooth (t) is present. By summing the counts of these decayed, missing, and/or filled teeth, we are able to assign a “dmft score” for each child. When total dmft > 0, there is evidence of early childhood caries. Severe early childhood caries adheres by the following guidelines, as explained in the Methods section: “sECC was defined based on the total number of decayed, missing or filled teeth (dmft) relative to the child’s age group: a dmft ≥1 in children under age 3, dmft ≥ 4 at age 3, dmft ≥ 5 at age 4, and dmft ≥ 6 at age 5 [11].”

7. Methods: A detailed explanation/ rationale for the choice of variables (such as electric lighting etc.) included in the analysis is needed?
Response: The survey tool used in this study has been adapted from standardized tools developed by the World Health Organization, as mentioned in the Methods (Lines 94-95), and survey questions were based on the Oral Health Surveys Basic Methods fifth edition. Additional socioeconomic status variables were also selected based on a literature review of socioeconomic determinants of nutrition and oral health. In the context of our study population—the majority of whom rely on subsistence farming for their daily livelihoods— it was not possible for our team to
inquire about each participant’s income. Instead, we opted to include variables that capture this information and serve as proxy measures for socioeconomic status as outlined in existing literature and research tools. (Please see
response #16 for further detail on the wealth index.)

8. Results: Include n for each variable in Table 1 and 2.
Response: We have updated the n-counts for each variable in Tables 1 and 2.

9. Results: Explain what is meant by ‘ever breastfed’?
Response: We have now defined the term at the first mention in the Results section.
• Lines 152-153: We have added the phrase “(any attempt at breastfeeding, even if only for a short time)” to define the term “ever breastfed.”

10. Results: Page 4, line 156: ‘baby bottle (12.9%) caused caries’ seems unclear?
Response: We agree that the wording here can be better clarified. We had intended for this to represent that only 12.9% of respondents recognized that use of the baby bottle can lead to tooth decay (i.e. baby bottle tooth decay from
regular exposure to sugary drinks and bottle use at night-time). We have now made the following changes to the description:
• Line 167: We have now updated the phrase to read “using the baby bottle (12.9%) can cause caries”

11. Discussion: Findings need a more thorough discussion and must be adequately supported by the literature.
Response: Based on Reviewer 3’s feedback, we have incorporated additional studies and text to more comprehensively cover all discussion points. Please see the following changes:
• Lines 276-277: We have now cited specific structural barriers to achieving healthy nutrition and positive oral health outcomes by adding the phrase “structural barriers— such as, but not limited to, the lack of affordable
nutritious foods and limited access to healthcare in rural areas”
• Lines 286-288: We have added further discussion of the healthy nutrition strategies by Pan American Health Organization with the sentence: “This recommendation builds upon the Pan American Health Organization’s
(PAHO) existing strategies for promoting healthy breastfeeding practices as part of their childhood obesity prevention efforts [40].” Here, we are also aligning the future policy recommendations based on our findings with the existing work currently being done by PAHO.
• Lines 303-307: We have added literature citing a socio-ecological model to support the key recommendations based on our findings: “Beyond further integration of maternal-child health and nutrition with oral health
interventions, these findings also support a socio-ecological approach for developing interventions at multiple levels—in the home, community, national, and international landscape—that address the socioeconomic and environmental factors influencing development of child undernutrition and sECC in rural El Salvador [44].”
• Lines 307-308: We have added literature explaining why our discussion focuses on school-based policies: “Past literature has indicated that the food environment at and surrounding schools is especially important for protecting children’s health [45].”
• Lines 314-316: We have cited additional challenges for feasible implementation of public health interventions, in similar limited-resource settings. The sentence “Prior research has noted that effective implementation of schoolbased obesity-prevention programs suffers when there is limited integration of the children’s families and school vendors in the program’s efforts [47]” describes potential barriers that regulations in El Salvador may also need to overcome in order to ensure greater enforcement of their nutrition policy changes. These implications are drawn from existing implementation research studies on school-based obesity prevention programs in Mexico, also conducted in the context of the global nutrition transition.
• Line 318-320: We have incorporated new literature supporting our discussion of SSB marketing, by identifying it as a driver of the nutrition transition in Latin America with the phrase: “drafting additional restrictions for the marketing of fast foods and SSB’s—a driver of the nutrition transition in Latin America [18, 44, 48, 49].”
• Lines 320-323: We have also considered El Salvador-specific barriers that may limit implementation of these nutrition and oral health strategies with this addition: “The feasibility of public health interventions in El Salvador is also limited by the presence of context-specific challenges, including gang violence threatening implementation of sustainable development; high vulnerability to natural disasters severely impacting infrastructure and agriculture; and limited integration of national health systems [50].” These barriers are cited by PAHO.

12. Discussion: This section is currently unorganised and incoherent.
Response: We have now included sub-headings to better organize the Discussion section. These sub-headings can be found at the following locations in the text:
• Line 237: “4.1. Sugar-sweetened Beverages (SSB’s), Bottle Feeding, and Adverse Health Outcomes”
• Line 259: “4.1 Association between Early Childhood Caries (ECC) and Undernutrition”
• Line 268: “4.3. Demographics and Socioeconomic Determinants”
• Line 279: “4.4 Recommendations”
• Line 327: “4.5 Study Limitations and Strengths”

13. Discussion: Several findings need clear discussion, supported with evidence and clarity, such as: WHY Children fed from baby bottles with SSB’s had 2.26 times increased odds of being more underweight?
Response: We have incorporated the following changes to better explain why consumption of SSB’s through the baby bottle may contribute to developing undernutrition. However, we also want to emphasize that because our study is
cross-sectional, we have only examined correlations and are unable to establish causality in our findings. We have acknowledged this as one limitation of our study, but our findings are strengthened by existing literature cited in the
Discussion that have also found SSB consumption as a risk factor for developing malnutrition and early childhood caries.
• Lines 254-258: We have added the following to better describe plausible mechanisms by which frequent SSB consumption through the baby bottle may result in increased odds for underweight: “Frequent consumption of sugary drinks might contribute to undernutrition through suppressing children’s appetite for nutritious foods [36]. There are multiple possible physiological pathways: undernutrition can arise from sECC causing chronic infection/inflammation and mouth pain that can interfere with appetite and chewing ability; or chronic
infection/inflammation can cause dysregulation of hepcidin and the hypothalamic-pituitary-adrenal hormonal axis.”

14. Discussion: WHAT is meant by ‘Healthy bottle-feeding practices’?
Response: As described in the Discussion, we are using the term ‘healthy bottle-feeding practices’ to refer to the recommendations outlined by the American Association of Pediatrics. These guidelines can be found in Lines 284-286
and are as follows: “clear guidelines for bottle-feeding when needed (including feeding babies only breastmilk or formula [not sugar-sweetened liquids] in the baby bottle, not putting the baby to bed with the bottle, and stopping the
bottle after 12 months of age).”

15. Discussion: Mothers’ oral health knowledge was not significantly associated with nutrition or oral health outcomes (OF ??) may be indicative of the structural health barriers (WHAT ARE THESE??) to ensuring good oral health in rural El Salvador
Response: We have now clarified that their knowledge was not associated with children’s nutrition/oral health outcomes and have specified which structural barriers are potentially limiting good oral health and nutrition outcomes in rural El Salvador
• Lines 275-278: We have added the following phrase to specify which structural barriers we are referencing: “Our finding that mothers’ oral health knowledge was not significantly associated with children’s nutrition or oral health outcomes may be indicative of structural health barriers—such as, but not limited to, the lack of affordable nutritious foods and limited access to healthcare in rural areas— to ensuring good nutrition and oral health in rural El Salvador [18].”

16. Discussion: Support with evidence “wood-fuel cooking as a proxy for low economic status” 
Response: As mentioned in the Discussion, existing research has identified wood-fuel cooking as an indicator of limited/low-resource settings (Lines 270-271). Further, the wealth index—a standard measure of household living standard constructed by the Demographic and Health Surveys Program (DHS)— also recognizes a household’s cooking fuel type as a key variable to consider when estimating a household’s relative wealth. We acknowledge that using wood-fuel cooking as a proxy for low economic status is not a perfect measure and disproportionately represents/only captures a certain segment of the population, and discuss our use of this, as compared to estimating household-level wealth through financial assets.

17. Discussion: No acknowledgement that the results could be potentially confounded by measures (give examples?) not captured in the dataset.
Response: We agree that the impact of additional measures can be better discussed. We have added the following measures not captured by our dataset that may be influencing our results 
• Lines 332-335: We have added the following sentence, supported by child undernutrition research in Latin America: “Undernutrition outcomes could also have been more accurately modeled had the survey captured details on consumption of healthy foods, child infectious disease history, maternal malnutrition history and practices, and micronutrient deficiency [52].”

18. Discussion: Paragraph on implications for future research and policy needs strengthening and discussion. 
Response: Section 4.4 Recommendations within the Discussion section outlines how future policy can incorporate the findings of this study. We have also incorporated new details on potential feasibility challenges for incorporating the
policy changes called for by our study. These are outlined below. To better clarify how future research can improve our findings, we have also incorporated the following:
• Lines 314-316: We have cited additional challenges for feasible implementation of public health interventions, in similar limited-resource settings. The sentence “Prior research has noted that effective implementation of schoolbased obesity-prevention programs suffers when there is limited integration of the children’s families and school vendors in the program’s efforts [47]” describes potential barriers that regulations in El Salvador may also need to overcome in order to ensure greater enforcement of their nutrition policy changes. These implications are drawn from existing implementation research studies on school-based obesity prevention programs in Mexico, also conducted in the context of the global nutrition transition.
• Lines 320-323: We have also considered El Salvador-specific barriers that may limit implementation of these nutrition and oral health strategies with this addition: “The feasibility of public health interventions in El Salvador is also limited by the presence of context-specific challenges, including gang violence threatening implementation of sustainable development; high vulnerability to natural disasters severely impacting infrastructure and agriculture; and limited integration of national health systems [50].” These barriers are cited by PAHO.
• Lines 348-349: We have added the following phrase to more specifically describe future research needed “on the effectiveness of interventions to reduce consumption of SSB’s in the baby bottle on the prevention of child undernutrition and sECC.”